# Does Placental Efficiency and Vascularization Affect Puppy Health? A Study in Boxer and Dobermann Dogs

**DOI:** 10.3390/ani14030423

**Published:** 2024-01-28

**Authors:** Alessia Gloria, Maria Cristina Veronesi, Alberto Contri

**Affiliations:** 1Department of Veterinary Medicine, University of Teramo, Località Piano D’Accio, 64100 Teramo, Italy; agloria@unite.it; 2Department of Veterinary Medicine and Animal Sciences, Università degli Studi di Milano, Via dell’Università, 26900 Lodi, Italy; maria.veronesi@unimi.it

**Keywords:** dog, puppy, placenta, placental efficiency, vascular density

## Abstract

**Simple Summary:**

Life inside the uterus is possible only thanks to the placenta, an essential structure that undergoes modifications to support the growth and development of the fetus. For this reason, the placenta is also described providing as a “diary of intrauterine life” and is a potential source of information regarding events that occur during gestation. Few studies, however, have considered the evaluation of this organ in the management of canine neonates. In the present study, the efficiency of the placenta was evaluated in 69 live puppies delivered by 15 large-breed bitches. The data reported in the present study show that in a subgroup of puppies, despite a normal placental weight and structure, reduced placental efficiency was identified. Although the findings should be confirmed in a larger population, the significantly increased risk of death in low-efficiency placental puppies reported in this study suggest the usefulness of this parameter in the evaluation of the puppy in the first 7 days after delivery.

**Abstract:**

Background: The anatomical and functional characteristics of the placenta influence the adaptive ability of the fetus to the extrauterine environment. Placental efficiency, measured as the gram of neonate produced by each gram of placenta, summarizes these characteristics. In the present study, placental efficiency and its impact on the 7-day postpartum life of the puppies were studied in canine large breeds. Methods: Placental efficiency (PE) was computed using chorioallantois weight (WPE) and surface (SPE) efficiency for puppies born from natural delivery or elective cesarean section. Capillary density was also histologically determined. Neonate viability was estimated by the APGAR score and the daily weight gain (DWG) was recorded on day 7 after delivery. Results: from 15 large-breed bitches, 69 live puppies were born by natural delivery (24 puppies) and elective cesarean section (45 puppies). Cluster analysis detected a group of neonates for which reduced placental efficiency (8 for the WPE, 9 for the SPE) was identified, despite a placental weight and surface within the mean and no difference in capillary density. In this group, the DWG was lower and the mortality within 7 days was higher. Conclusions: the results suggest that placental efficiency could be an additional tool for the evaluation of a puppy in the first 7 days after delivery.

## 1. Introduction

The placenta is defined as the interactional structure, made by the apposition or fusion of fetal membranes and maternal endometrium [1], between the developing fetuses and the bitch. The mammalian placenta substitutes the fetal lung, gut, kidney, and liver during intrauterine life, providing gas exchanges, nutrients, and hormones [2].

Based on the morphology of the placental attachment to the endometrium, the canine placenta is defined as zonary, with a circumferential adhesion band [3]. The adhesion band is bordered by two circle-shaped hemophagous areas, which are green-pigmented, and extensive areas of smooth chorioallantois, the paraplacenta [4,5]. Based on the number of tissues separating the fetal and maternal circulation, the canine placenta is also defined as endotheliochorial due to the trophoblast invasion of the maternal endometrium until the surrounding of the maternal capillaries [6].

Due to their crucial supportive role, placental morphology and function drive fetal development during in utero growth and could affect short- and long-term postnatal health. Several studies suggested a direct link between placental dysfunction or lack of nutrient intake during fetal life and the possibility of developing pathologies such as chronic diseases and cancer during adult life, or having a shorter lifespan [7,8,9]. In polytocous species, such as dogs [10], several placentas are distributed inside the uterine horns form to support fetuses in development. The supportive ability of the placenta appeared, however, variable, since in large litters there may be fetuses of different sizes with placentas that also have different surfaces and weights. Low-birthweight puppies, in a large population study, were found to be at higher risk of death [11]. Although the reasons for the low weight at birth, likely due to an intrauterine growth restriction (IUGR) [12], are not yet clearly elucidated, the modulatory and supportive role of the placenta appears crucial in this process. As a result, placental efficiency, defined as grams of the fetus compared to grams of the placenta [13], was considered a relevant parameter in the neonate assessment (for a review, [14]). In a study in mice, it was found that differences existing in fetal weight in early gestation became progressively less significant with pregnancy progression, paralleling with some morphological and functional modifications of the corresponding placentas. The authors concluded that placentas can modify their morphology/function to normalize the neonate birthweight [15]. However, the opposite could also be true, namely, that the inability of the placenta to adapt leads to alterations in fetus weight, placenta weight, or both [12]. Different studies on polytocous species have demonstrated how placental efficiency can vary by up to 100% within a litter [13,16,17].

Due to the preparatory role of the placenta on fetal and neonate well-being, a functioning placenta is particularly important in the dog, in which the principal aim of the breeder is to maximize the number of weaned puppies. Evaluation of the placenta represents a complementary to the clinical evaluation of the neonate. Recently, studies showed the usefulness of morphological and histological assessment of fetal adnexa in the canine puppy, showing a relationship between these values and the puppy’s vitality, birthweight, and postnatal growth [18,19]. The placenta expelled after delivery or extracted during caesarean section consists almost totally of fetal membranes with marginal contamination of maternal decidual cells [2].

Despite the pivotal role of the placenta in the development of the fetus and puppy, only limited attention is devoted to detecting alterations suggestive of adverse effects threatening the health of young dogs. To the authors’ knowledge, no studies have considered placental efficiency as a key factor in the puppy health of large dog breeds in the first week of life. The present study aimed to explore the correlations between placental characteristics, placental efficiency, and neonatal characteristics in a large-size breed, detecting whether signs of dysfunction or of the possible adaptation of this organ supported final fetal development.

## 2. Materials and Methods

### 2.1. Animals

The study was conducted on 15 bitches of known fertility aged between 2 and 5 years and weighing 28 to 35 kg. The breeds considered in this study were Boxer (n = 12) and Dobermann (n = 3). Animals were monitored during the pregnancy for controlled puppy vitality by ultrasound evaluation at the Hospital of the University of Veterinary Medicine of Teramo, Italy. All the dogs included in the study were managed according to the Italian legislation concerning animal care (DL n.116, 27 January 1992), and consent was obtained from the owner for the use of data from their animals and placentas.

### 2.2. Reproductive Management

At the first visit, each bitch underwent a general and specific clinical examination of the female reproductive system. Estrous phase and ovulation were monitored via colpocitology and blood progesterone concentration to define the optimal moment of natural breeding or artificial insemination with fresh semen. A proportion of epithelial cells greater than 90% was indicative of estrus, and progesteronemia (chemiluminescence method, [20]) between 6 and 10 ng/mL was considered indicative of ovulation [21,22]. The bitches underwent natural mating or artificial insemination with fresh semen at 1 and 3 days after ovulation. Pregnancy was diagnosed ultrasonographically at day 20 after ovulation with an Esaote MyLab Seven ultrasound (Esaote spa, Genoa—Italy) equipped with a multifrequency micro-convex probe (4–9 Mhz) and confirmed at day 25, detecting the heart activity as fast alternate between hypoechoic and hyperechoic dots inside the embryo mass. The nutrition status of the bitches was considered at the first visit and then every two weeks until parturition. The evaluation was based on the 9-point score described in the World Small Animal Veterinary Association (WSAVA) nutrition assessment guidelines [23].

Starting from day 61 after ovulation, the bitches were clinically monitored and daily progesteronemia was measured until parturition. In animals designed for elective caesarean section, the surgical procedures were performed at 62 days after ovulation, with a value of blood progesterone < 1 ng/mL and a fetal heart rate < 180 bpm.

### 2.3. Birth Management and Data Collection

Bitches with dystocic parturition or requiring emergency caesarean section were excluded from the study. In normal parturition bitches, the parturition was supervised, and the order of expulsion was recorded. For elective caesarean section, the anesthetic protocol used involved the use of fentanyl in premedication at a dosage of 4 µg/kg. Induction was carried out after 5 min with propofol at a maximum dosage of 4 mg/kg intravenously. General anesthesia was maintained with 2% isoflurane and fentanyl in CRI at 0.05 µg/kg/minute until extraction of all pups. Cefazolin (20 mg/kg) was administered intravenously soon before the surgical procedure, then subcutaneously BID for 7 days.

### 2.4. Puppy Management and Data Collection

In the case of spontaneous birth, once the puppy has been expelled, the placenta is removed, avoiding ingestion by the mother. In both natural delivery and cesarean section, the puppies were moved to a warm area and received neonatal care consisting of the removal of mucous and fluids (also using manual aspirators) and thoracic massage to stimulate breathing. The puppies were accurately dried with clean towels. Soon after natural birth or caesarean section and the conventional nursing procedures, each puppy was examined to check congenital abnormalities and record the gender. Neonatal viability is assessed using the APGAR method [24] 5 min after birth, and subsequently 60 and 120 min after birth. The APGAR score is based on the following clinical evaluations: appearance of the mucus membrane (A), rated with 2 when pink, 1 when pale pink, and 0 when cyanotic or white; heart rate (P), scored 2 when >220 bpm, 1 when between 180 and 220, and 0 when <180; grimace (G) as the reflex irritability, which was evocated by a gentle compression of the tip of the paw and was scored 2 with crying and immediate retraction, 1 with weak retraction and vocalization, and 0 when no reaction was induced; activity (A), estimated by the observation of the strength of autonomous movements of the puppy, scored as 2 with strong movements, 1 with mild movements, and 0 with weak or no movements; and respiration (R), scored on the basis of respiratory rate and spontaneous crying, recorded as 2 when clear crying was associated with rpm >15, 1 when mild crying was associated with rpm between 6 and 15, and 0 when no crying with rpm < 6 were present. The APGAR score was computed on the sum of the scores of each parameter. The puppies were subsequently weighed using a precision balance (CP6201—Sartorius AG, Gottingen, Germany) before the first feeding and placed in a quiet environment with their mother, warmed by the presence of an infrared lamp. The litter size, as the number of puppies delivered by each bitch, was recorded at the end of parturition.

The weight at d0 and d7 or after death was measured and the difference in weight was subdivided by the number of days concerned, calculating the daily weight growth (DWG). The puppy mortality rate was verified on day 7 after delivery.

### 2.5. Placenta Evaluation

The placenta of each puppy was ordered for birth (natural delivery) or extraction (caesarean section). The morphometric evaluations took into consideration exclusively the chorioallantois (CA); thus, the other fetal membranes were removed before the measurements. The weight of the CA (CAW) was determined by the precision balance CP6201. The CA was cut transversally to be a linear conformation, carefully distended on a grid paper, and digitally photographed. The image obtained was analyzed using ImageJ v 1.52a (http://imagej.nih.gov/ij/ on 30 April 2018), calculating objectively the area based on the automatically detected outline (Figure 1) as the region of interest (ROI). The CA surface (CAS) was automatically calculated by the software.

The weight placental efficiency (WPE) was calculated, according to studies in mammals [12,13], using birthweight and placental weight ratio, indicative of the gram of fetus produced per gram of CA. Furthermore, the surface placental efficiency (SPE) was also calculated as the gram of fetus produced per cm square of CA surface.

Capillary density was estimated as previously described, with few modifications [18]. After macroscopic analysis, 3 transversal sections (about 2 cm in length) were prepared in different CA regions and fixed in Bouin’s solution for 24 h, then dehydrated in increasing concentrations of ethanol and embedded in paraffin. From each section, 3 non-consecutive slices 4–6 µm thick were transferred onto slides and stained with hematoxylin-eosin (HE). The sections were analyzed using the Olympus BX51 microscope (Olympus Lifescience, Tokyo, Japan) with a 10× objective (100× magnification) and 5 randomly selected fields were digitalized. In post-editing, a grid with squares of 100 µm was superimposed on the digitized image (Figure 2). The number of capillaries in 10 randomly selected squares for each field was counted, and the mean capillary density (CD—capillaries/mm^2^) was calculated using the following formula:CD = mean square capillaries × 100

### 2.6. Statistical Analysis

The values for puppy birthweight, CAW, and CAS were normalized within each litter by the calculation of the absolute deviation as the difference between the value and the mean of the litter. The differences in birthweight and CA morphometric parameters (CAW and CAS) were compared according to the different variables using the general linear model (GLM) based on univariate ANOVA, in which the breed of the bitch, the type of birth (natural birth or caesarean section), the sex of the puppy (male and female), the litter size, and the order of parturition/extraction were considered fixed variables. When appropriate, the Scheffè test was used for the post hoc analysis. Similarly, the WPE and the SPE were compared in the different groups (natural birth or caesarean section; sex of the puppy, and the order of parturition/extraction) using the GLM and the Scheffè test. The WPE and SPE of live puppies, excluding the data from stillborn puppies and those with congenital malformations, were used to classify groups with significant differences using the two-step cluster analysis, based on Schwarz’s Bayesian Criterion. Placental morphology parameter values and absolute deviations, and DWG were compared between clusters using GLM. The odds ratio for the 7-day mortality in the different clusters was calculated. Finally, correlations between puppy weight, CA morphometric parameters, WPE, SPE, CD, puppies’ APGAR5, APGAR60, APGAR 120, and DWG were assessed using the Spearman’s Rho test. In all the cases, the significance level was set at *p* < 0.05. Statistical evaluations were performed using SPSS 17.0 software (SPSS Inc., Chicago, IL, USA).

## 3. Results

In this study, 75 puppies from 15 bitches were considered, with a mean litter number of 6.1 ± 2.7 puppies. The mean pregnancy length, calculated from the estimated ovulation to the parturition, was 63.3 ± 0.7 days. Of the bitches considered in the study, six had a natural birth and nine underwent elective caesarean section. The mean weight of the bitches before parturition was 31.7 ± 3.8 kg. The BCS estimated in the bitches included in this study was 4.6 ± 0.3 (range 4–5) and was not modified during pregnancy.

The 75 puppies delivered comprised 41 females and 34 males. In bitches with a natural delivery, 28 puppies were delivered, with a survival rate of 85.7% (24/28 puppies): 4 puppies were expelled without cardiac activity, without alterations in the uterine/abdominal contractions of the bitch. The survival of puppies born by caesarean section was 95.6% (45/47), since 2 fetuses were extracted without heart activity. Of the remaining 69 puppies, 6 (8.69%) showed severe congenital malformations (2 palatoschisis, 2 gnatho-palatoschisis, 1 imperforate anus, 1 tracheal atresia) and were excluded from the analysis. The mean weight of the 63 puppies was 449.1 ± 55.9 g, with significantly higher weight for males (462.4 ± 31.3 g) compared to female puppies (438.2 ± 26.6 g; *p* < 0.05). The APGAR at 5 min was significantly higher in puppies born after natural delivery compared with those born after caesarean section (*p* < 0.05), while the differences were not significant at 60 and 120 min (Table 1).

The mean CAW, CAS, and WPE values in the study population were 49.7 ± 7.5 g, 92.5 ± 17.7 cm^2^, and 9.6 ± 1.9, with no differences between male and female fetuses (Table 2). The SPE calculated in the mean population was 5.2 ± 1.2, with values higher in males compared with females (Table 2; *p* < 0.05). The litter size was not significant for any parameter considered (*p* > 0.05). Compared with live puppies, similar values were found for birthweight, CAW, CAS, WPE, and SPE in stillborn puppies (*p* > 0.05) and in puppies showing congenital abnormalities (*p* > 0.05).

Clusterization performed on the WPE using two-step cluster analysis showed the presence of two clusters. One subgroup of 8 CAs (12.7%) showed a significantly lower WPE (low-weight cluster—LWCl; 8.6 ± 1.3) compared with the other 55 (normal-weight cluster—NWCl; 10.8 ± 1.8). Clusterization performed on the SPE showed a similar trend, with lower values in a subgroup of 9 CAs (14.3%; low surface cluster—LSCl) compared with the other 54 CAs (normal surface cluster—NSCl; 4.3 ± 0.5 vs. 5.6 ± 1.1). The absolute deviations of birthweight, CAW, and CAS were not significantly different between clusters, and neither were there significant differences when clusterization was based on the WPE or SPE (*p* > 0.05).

Three puppies out of 63 died within the observation period of 7 days (4.8%), 2 in the LWCl/LSCl and 1 in the NWCl/NSCl. The odds ratio for survival at day 7 postpartum was significantly different in both the WPE (OR = 18; 95% CI = 1.4134–229.23; *p* < 0.05) and SPE (OR = 15.1, 95% CI = 1.2104–189.45; *p* < 0.05) clusters.

The mean CD was 267.7 ± 61.9/mm^2^, without differences in the CA between male and female puppies. The CD in the LWCl was similar compared with that in the NWCl (261 ± 46.4/mm^2^ vs. 278.6 ± 41.6/mm^2^). Similarly, the CD in the LSCl was similar compared with the NSCl (252 ± 37.8/mm^2^ vs. 274.1 ± 40.7/mm^2^; *p* < 0.05).

The mean DWG in this study was 53.9 ± 12.1 g, with no differences between the fetal gender or the delivery type. The DWG in the LWCl was significantly lower than in the NWCl (26.3 ± 12.3 g vs. 58.2 ± 2.3 g, respectively (*p* < 0.05). Similarly, significant differences were found between the DWG in the LSCl (25 ± 11.9 g) compared with the NSCl (58.1 ± 11.9; *p* < 0.05).

The parameters recorded in the present study were similar in the different breeds (*p* > 0.05).

In the general study population, the weight of the puppy was significantly correlated to the CAW (r = 0.482; *p* < 0.05) and CAS (r = 0.528; *p* < 0.05). The correlations were made more significant by removing from the computation the puppies within the LWCl (r = 0.816; *p* < 0.01) and the LSCl (r = 0.874; *p* < 0.01). In turn, a significant correlation was recorded between the CAW and CAS (r = 0.830; *p* < 0.01). As shown in Figure 3, significant correlations were found between the DWG and the CAW (r = 0.844; *p* < 0.01) in the NWCl but not in the total population (r = 0.484; *p* > 0.05). Similarly, the DWG was positively correlated to the CAS (r = 0.843; *p* < 0.01) in the NSCl (Figure 3), but not in the total population (r = 0.476; *p* > 0.05). None of the other parameters considered in the present study showed significant correlations.

## 4. Discussion

In the present study, placental efficiency in dogs was estimated, verifying the implications of population-divergent PE on the postnatal life of puppies. Other visible factors affecting the survival of the fetuses resulted in the exclusion of puppies and their CA from the statistical analysis. The morphology and size of the puppies included in the present study were similar to the data presented in the literature of these large-size breeds [25]. The neonatal survival rate recorded after cesarean section is 96%, near to the values reported in the literature (92% by [26]; 95.1% by [27]; 99% by [28] and [29]). The congenital malformation rate recorded in this study was 8.6%, lower than the value reported for the breed most represented in the present study of about 14.9% [30] and similar to that reported in canine heterogeneous breeds [31].

Cesarean section affected the APGAR score at 5 min, which was significantly lower than after natural delivery. This finding could be related to the anesthetic protocol since similar values for the APGAR score were detected in previous studies [27,28]. The placenta offers a variably effective barrier to anesthetic drugs, which, in general, can diffuse in the fetal circulation causing different degrees of depression [32]. The metabolism of the puppy progressively eliminates these drugs [30], resulting in the rapid implementation of fetal viability, demonstrated by APGAR scores at 60 and 120 min similar to those of vaginal delivery. A similar trend was previously reported in other studies [27,33], suggesting the elimination of the molecules used for the surgical procedures likely is responsible for this finding.

Within a litter, placental weight and surface were variable between different puppies. This is likely the reason why the litter size was not found to be a significant variability source. Litter size was found to affect placental efficiency, with higher values in large compared with small litters [18]. In that study, discrimination was based on a comparison with the mean litter size reported for a specific breed. The litter size recorded in the present study was within the range reported in the literature [34], reducing the effect of this component on the WPE or SPE recorded. No significant differences were found for the use of WPE and SPE, as demonstrated by the correlation between these parameters in the normal population, suggesting that these parameters could both be used effectively. The evaluation of the placenta surface available for fetal exchange was found effective in estimating placental efficiency [35,36], but the procedure for its calculation is articulated, less direct, and requires several passages, resulting in it being less frequently applied in practice.

Birth is an event that generates a set of changes and leads the individual to move from an intrauterine to extrauterine life. Pregnancy is crucial to preparing the puppy for an independent extrauterine life, and the placenta assumes a pivotal role in driving this development. A reduction in placental efficiency was associated, in humans, with increased risks of raised blood pressure in the young [37], hypertension [38], and cardiac disease in adulthood [39]. Increasing evidence suggests that the placenta can adapt to fetal requirements, modifying its anatomical and functional characteristics to support normal fetal growth [12,15,40]. In humans, an increased placental efficiency was detected in smaller-than-normal placentas, with 20% more fetal mass supported by a placenta of less than 300 g compared with a placenta of more than 500 g [41]. Thus, the WPE and SPE, rather than the placental weight or surface, appeared to be an appropriate method to verify placental function in dogs. Studies in rodents showed that experimentally induced modifications in maternal nutrition, blood oxygenation, and uterine blood flow could all affect the placenta weight. Of these factors, oxygen restriction [42,43,44] and uterine blood restriction [45,46,47] reduced the placental efficiency, while overnutrition especially in fat [48] or nutrient restriction [42,49] resulted in an increase in placental efficiency.

The role of the vasculature in placental adaptation and, in turn, in placental efficiency is controversial. In pigs, a smaller but more efficient placenta within a litter showed an increase in capillary density [50]. On the other hand, in nulliparous ewes, increased efficiency of the placenta was associated with a reduction in vasculature and VEGF expression [51,52]. In the present study, no differences were found between the CD in the LWCl compared with the NWCl and in the LSCl compared with the NSCl, according to findings in rats [53]. The inconsistent relationship between placental efficiency and capillary density should be related to the differences in placental morphology and physiology. Differences exist in placental shape and distribution, with diffuse (swine and equine), cotyledonary (ruminants), zonary (carnivores), or discoid (human and rodents), internal structure, interhemal barrier, with epitheliochorial (with intact uterine epithelium, i.e., horse, ruminants), endotheliochorial (invasion of the endometrium and contact of the trophoblast with the maternal capillaries, i.e., carnivores and bats), or hemochorial placentation (with no maternal tissues in the interhemal barrier, i.e., mouse and guinea pigs) [3]. These structural and functional differences implied a different vascular organization, making it difficult to compare the different species directly. Moreover, intra-specific differences were reported in pigs and sheep, where an increased capillary density was associated with more prolific breeds [12].

In the present study, a group of puppies showed a CA weight or surface similar to the mean study population but resulted in low-weight puppies at birth, suggesting a reduced ability of this structure to support fetal development or the inability of the fetus to stimulate placental growth and adaptation. On the other hand, this finding suggested that placental weight could not be the sole parameter evaluated at birth in this species. On the other hand, in a previous study on the effect of birthweight on neonatal weight growth and risk of mortality in dogs, it was found to be a biphasic effect of the low birthweight condition, with puppies showing a reduced growth rate compared with normal-weight puppies between day 0 and 2, but an increased between day 2 to 7. The same authors reported higher mortality in the low birthweight group between day 0 to 2, with a drastic reduction in the period from day 0 to day 7, detecting significantly higher mortality in low birthweight puppies with low growth rates [54]. Although no data were reported in that study regarding placental evaluation, the data reported in that study found that differences were present in the group of low birthweight puppies and that the combination of low birthweight and low early growth rate should be considered to assess correctly the at-risk puppies. In the present study, a different approach was followed, namely, the comparison of puppies classified based on placental efficiency. The aggregate neonatal mortality was similar [19] or lower than in other studies [30,55], but the puppies within the LWCl and LSCl showed an increased risk of death within 7 days of delivery. This evidence, corroborated by the presence of one or few fetuses with low placental efficiency in litters in which the majority of the fetuses showed normal placental efficiency, suggested that the factor responsible for the reduction in the fetal growth is likely fetal or placental rather than maternal. The conclusions of the present study were consistent with the previous finding by Mugnier and co-workers [54], suggesting that a fetal or placental defective function, demonstrated by reduced placental efficiency, impacted the birthweight and the puppy growth rate in the first days of extrauterine life. Thus, placental efficiency based on weight represents a useful and preventive parameter for the evaluation of the neonate in dogs, allowing the identification of puppies that could, in the first weeks of life, be at risk and require more attention and support. The basic technological requirements, the low clinical competencies necessary to estimate placental efficiency, and its objective nature make this parameter easily performed in field conditions by the breeder, in addition to other tested practices, such as the modified APGAR score, the expulsion time, and the birthweight [56]. In a previous study on the histology of the canine placenta at term, some placental disorders, such as necrosis or degenerative calcification, were associated with a higher risk of death for the puppy in the first week of life [18]. In the present study, such histological features were occasionally detected, and in some cases were also identified in the group of normal WPE/SPE, suggesting that this is not the case for the population involved in the present study. Although the individual diagnosis of the reason for death in the animals included was not possible, it is likely that the factors underlying this higher risk reside in the fetus/puppy. The question of whether the reason for the death is a primary defect of the puppy or an alteration that predisposes the onset of common neonatal problems remains to be elucidated in specific studies.

## 5. Conclusions

In the present study, placental efficiency, estimated both using the CA weight and surface, allowed the detection of a group of puppies with low placental efficiency, independently of birthweight. In this group, the postnatal growth rate was lower and the mortality was higher, suggesting that placental efficiency could be a useful parameter in the evaluation of the canine neonate. It is important to remark, however, that the sample size considered in the present study was limited; thus, further studies are required to confirm the role of placental efficiency in the prediction of puppies at risk in the neonatal period.

The basic technological requirements, the low clinical competencies necessary to estimate placental efficiency, and its objective nature make this parameter easily performed in field conditions by the breeder, in addition to other tested practices, such as the modified APGAR score, the expulsion time, and the birthweight.

## Figures and Tables

**Figure 1 animals-14-00423-f001:**
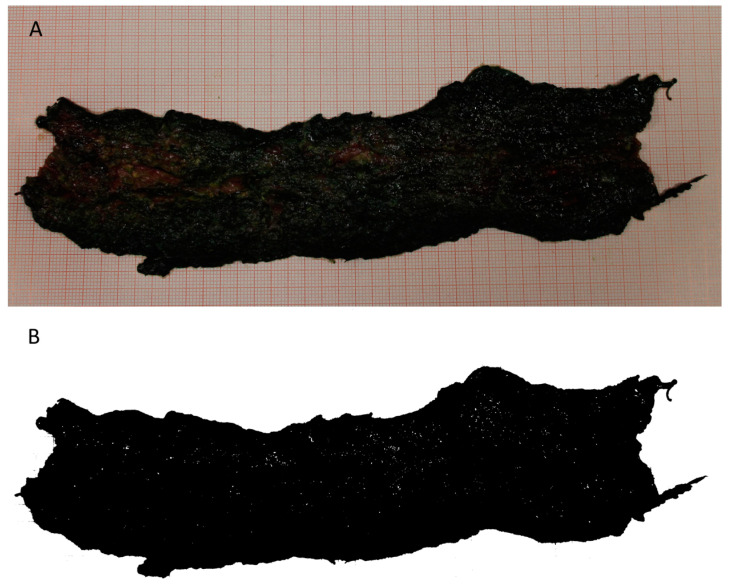
Representative figure of the canine distended chorioallantois (**A**) and its digitalization using ImageJ (**B**).

**Figure 2 animals-14-00423-f002:**
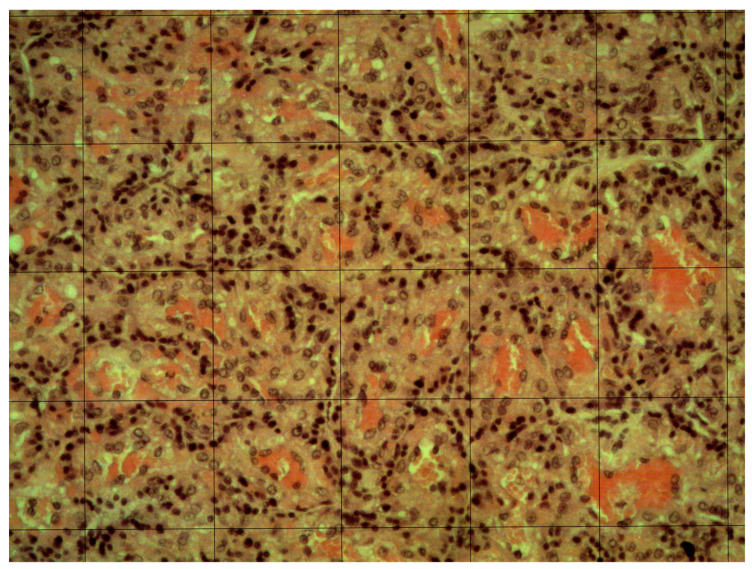
Representative image of canine chorioallantois histologic field, after staining with hematoxylin-eosin, with the superimposed 100-µm square grid (100× magnification).

**Figure 3 animals-14-00423-f003:**
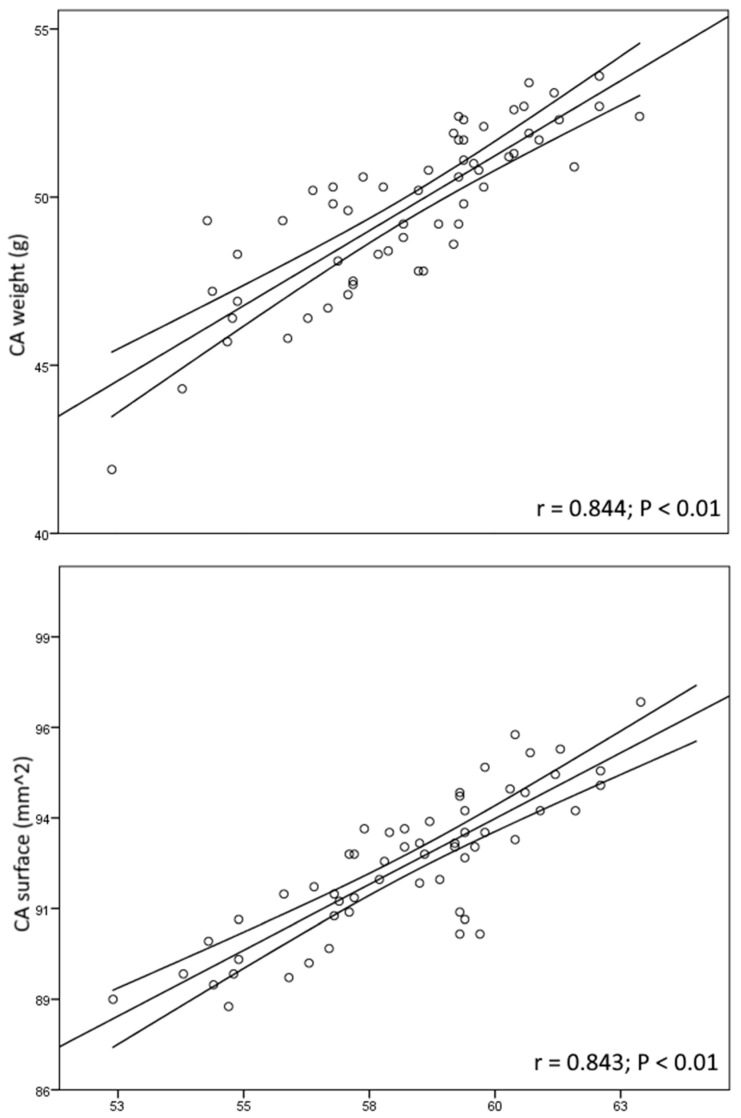
Plot distribution graphs of the correlations between chorioallantois (CA) weight and daily weight gain (DWG) (upper graph) and between chorioallantois surface and DWG (lower graph) in the cluster with normal placental efficiency. The solid straight line represents the fit line, and the solid curved represents the confidence intervals.

**Table 1 animals-14-00423-t001:** APGAR score recorded at 5, 60, and 120 minutes in puppies born after natural delivery or caesarean section.

	APGAR Score	
	Natural Delivery	Caesarean Section
5 min	7.8 ± 1.6 ^a^	6.4 ± 1.2 ^b^
60 min	8.9 ± 1.1 ^a^	8.6 ± 1.4 ^a^
120 min	9.1 ± 0.9 ^a^	9 ± 1.2 ^a^

In the same row, values with different letters in superscript differ significantly (*p* < 0.05).

**Table 2 animals-14-00423-t002:** Measurable placental attributes (chorioallantois weight—CAW; chorioallantois surface—CAS; placental efficiency estimated on weight—WPE; placental efficiency estimated on surface—SPE) in male and female puppies.

	Mean Population	Male Puppies	Female Puppies
CAW (g)	49.7 ± 7.5	51.2 ± 5.2 ^a^	46.4 ± 5.6 ^a^
CAS (cm^2^)	92.5 ± 17.7	95.6 ± 13.9 ^a^	90.3 ± 9.4 ^a^
WPE	9.6 ± 1.9	9.8 ± 1.7 ^a^	9.4 ± 1.4 ^a^
SPE	5.2 ± 1.2	5.2 ± 0.9 ^a^	4.9 ± 1 ^b^

The values of the same parameter with different letters between male and female puppies differ significantly (*p* < 0.05).

## Data Availability

The data are available from the authors, and can be obtained upon reasonable request.

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
