# Peer review of "Does Placental Efficiency and Vascularization Affect Puppy Health? A Study in Boxer and Dobermann Dogs"

_animals, 2024, doi:10.3390/ani14030423_

Round 1

Reviewer 1 Report

Comments and Suggestions for Authors

The authors investigated the associative effects of both anatomical and functional impact of the placenta on neonatal pup viability in canine large breeds. Two different common large breed gestating females were utilized for the experiment.  The outcomes of the study may not include groundbreaking evidence, but the results are an important contribution to a growing body of evidence investigating underlying causes of canine neonate viability.  The data is informative but there are weaknesses of the manuscript that should be edited to strengthen presentation and impact of the data:

1.     Line 25 and 125: define APGAR and scoring system

2.     Statistics section: Effect of breed should be included in the statistical analysis and reported.

3.     Results:

a.     Were congenital abnormalities correlated with placental efficiency, i.e., was the absence of a link investigated prior to exclusion from the study?

b.     Lines 185-207: A table(s) needs to be included to better clarify the results of puppy parameters. This would better highlight the significant differences of the statistical comparisons.

c.     Figure 3 needs to include the significant statistical probability and r values.

Author Response

The Authors would like to express their gratitude for the time and the constructive comments that will improve significantly the quality of our manuscript.

Here a point-by-point answer to each comment, and the requested changes done to the manuscript.

  1. Line 25 and 125: define APGAR and scoring system

R: The APGAR score was defined in the materials and methods, while in the abstract the length limitations impeded explanation in extenso. APGAR is a consolidated clinical evaluation in the neonate, and it should not be considered as an abbreviation.

  1. Statistics section: Effect of breed should be included in the statistical analysis and reported.

R: As suggested, the breed was included in the model, but it did not show a significant effect. We reported it in the materials and methods, and in the result sections.

  1. Results:
  2. Were congenital abnormalities correlated with placental efficiency, i.e., was the absence of a link investigated prior to exclusion from the study?

R We tested if the placental efficiency or birthweight were different between normal and abnormal puppies, and we found no statistical differences. We inserted this information in the results section.

  1. Lines 185-207: A table(s) needs to be included to better clarify the results of puppy parameters. This would better highlight the significant differences of the statistical comparisons.

R: As suggested by the Reviewer 1, we introduced Table 1 to summarize the parameters recorded in the puppies. The definition of the animals that met the criteria of inclusion should not be presented by a Table, because of the risk of misunderstandings.

  1. Figure 3 needs to include the significant statistical probability and r values.

R: the correlation coefficient (r) and the probability were inserted in the Figure 3.

Reviewer 2 Report

Comments and Suggestions for Authors

The article "Does placental efficiency and vascularization affect puppy
health? A study in large-size breeds. " submitted by Gloria et al. aimed to explore if analysis of placenta could be useful to detect puppies at risk of later early mortality.

The interest for the understanding of the neonatal puppy mortality is increasing since few years and the hypothesis, raised by the authors, that the placenta efficiency could play a role, is interesting.

Nevertheless, despite a good methodology used by the experimentators, the results section is quite unclear and the discussion part is missing a lot of important points.

Please find below some suggestions of improvements for the authors:

 - Did the BCS of the females have been recorded before parturition? It has been shown in other studies that overweight or obesity of the mother can impact offspring health. This parameter should have been taken into account in your analysis.

 - The number of severe congenital abnormalities seems very high (8.7%). Is it common in these breeds? Did such a rate has already been observed in previous litters of these dams?

 - Presentation of the results and statistics in a table would be much more convenient for the understanding by the reader. 

- Only 3 puppies died at Day 7, not enough for any relevant and consistent conclusion, even if statistics are significant. This point is not discussed in the "Discussion" section. One more or one less died and the outcomes would be totally different. The data are too preliminary and weak to write a sentence as "A significantly increased risk of death was present in low-efficient placental puppies, suggesting the usefulness of this parameter to detect canine neonates at risk in large breeds." in the summary. This must be confirmed with larger sample size before claiming this.

- It could have been interesting to show if data of stillborn puppies or puppies presenting congenital abnormalities were different of the others alive puppies.

- It's a pity that no comparison of the usefulness of these results to detect the puppies at risk of early mortality has been done in comparison of other existing methods: low birth weight, low early growth rate… etc.

- There is no discussion on how the knowledge presented in this paper could be applicable for breeders in comparison of  other existing methods.

- A real conclusion for this paper with perspective is lacking

- Line 41-43 : Sentence unclear, a word or dot is missing

Comments on the Quality of English Language

No comments

Author Response

The authors would like to thank the reviewer for the time spent in evaluating our manuscript.

Here we reported a point-by-point elucidation of the changes we performed, explaining their rationale.

- Did the BCS of the females have been recorded before parturition? It has been shown in other studies that overweight or obesity of the mother can impact offspring health. This parameter should have been taken into account in your analysis.

R: As correctly noted by the reviewer, the nutrition state could have a role in the progression of the pregnancy, affecting the fetus's development. In the present study, all the animals were from breeders or private owners, skilled with canine pregnancy, so the BCS was uniform in all the bitches included in the study, thus we did not consider it as a factor of variability. On the other hand, this information was not present in the manuscript and we inserted it in the text.

- The number of severe congenital abnormalities seems very high (8.7%). Is it common in these breeds? Did such a rate has already been observed in previous litters of these dams?

R: The rate of congenital malformation present is high, but it is similar to that reported in the literature in other brachycephalic and non-brachycephalic breeds. We inserted the sentence on this matter in the discussion, with the relative reference. The data are in line with our experience in these breeds, but the rate is recorded only on a limited number of animals belonging to these breed population, and we could not have direct data on all the population, but only anecdotal and referred. Since we are not sure about quantitative data, we preferred to avoid inserting such information in the text.

 - Presentation of the results and statistics in a table would be much more convenient for the understanding by the reader.

R: we inserted two tables (Tables 1 and 2) to summarize some descriptive results, with differences between APGAR score in natural delivery or caesarean section, or the placental parameters in the mean population, males and females, with appropriate statistical analysis.

- Only 3 puppies died at Day 7, not enough for any relevant and consistent conclusion, even if statistics are significant. This point is not discussed in the "Discussion" section. One more or one less died and the outcomes would be totally different. The data are too preliminary and weak to write a sentence as "A significantly increased risk of death was present in low-efficient placental puppies, suggesting the usefulness of this parameter to detect canine neonates at risk in large breeds." in the summary. This must be confirmed with a larger sample size before claiming this.

R: We are aware that the low number of divergent data could not be enough to have consistent results, on the other hand, we think our data could draw attention to the evaluation of the fetal membranes at parturition, and the role of the placental efficiency in the assessment of the neonate. Accordingly, with the suggestion of the reviewer, we modified the sentence specified, including a comment on the necessity of a confirmation on a large sample in the conclusion.

- It could have been interesting to show if data of stillborn puppies or puppies presenting congenital abnormalities were different of the others alive puppies.

R: We tested if the placental efficiency or birthweight were different between normal and abnormal puppies, and we found no statistical differences. We inserted this information in the results section.

- It's a pity that no comparison of the usefulness of these results to detect the puppies at risk of early mortality has been done in comparison of other existing methods: low birth weight, low early growth rate… etc.

R: the comment of the reviewer is of interest, suggesting the possibility to compare the different methods in their ability to detect at-risk puppies. The population number included in this study, however, appeared not sufficient to support this more complex statistical evaluation, but this suggestion could be surely considered in future study expanding the samples size.

- There is no discussion on how the knowledge presented in this paper could be applicable for breeders in comparison of other existing methods.

R: we inserted some sentences in the discussion and in the new conclusion paragraph to underline the useful of this parameter in addition to other established evaluations that could be performed in field conditions.

- A real conclusion for this paper with perspective is lacking

R: a conclusion section was inserted in the manuscript.

- Line 41-43 : Sentence unclear, a word or dot is missing

R: the sentence was modified, such as several parts of the introduction.

Reviewer 3 Report

Comments and Suggestions for Authors

The manuscript is interesting and relevant to those interested in the subject. I have made specific comments in the attached pdf.

All acronyms employed need to be explained when they are first mentioned. The lack of an explanation makes the in the Materials and Methods, Results and Discussion sections confusing and difficult to follow. Please remedy this.

As a Conclusions section is compulsory for this journal please add one. Please also make recommendations with regard to using placental efficiency as a parameter to assess the risk of puppy death.

Comments on the Quality of English Language

The Introduction section has long confusing sentences and a few factual and language mistakes. As such, it needs to be revised and rephrased so that the text can become correct and easy to understand.

Author Response

Authors would like to thank the reviewer for the time spent on our manuscript. The comments were all considered and the manuscript was modified accordingly. The changes improved the quality of our manuscript, and we hope now the manuscript could be clearer for the readers.

L42. Please make this be a separate sentence. It is also difficult to understand so please rephrase it.

L45-48. Please divide this in at least two sentences.

L54-57. Please divide this in at least two sentences.

L60-64. The point the authors are trying to make is difficult to understand. Please rephrase.

L66. Please rephrase this to make it factually correct and its meaning easy to understand.

R: we revised extensively this part of the introduction, following the recommendations of the Reviewers 3 and 4. All the aspects pointed out were considered and properly modified.

L73. Please replace with "weaned puppies".

R: the text was modified as kindly suggested by the reviewer.

L134. Please explain what this acronym means.

R: The acronym was explained in extenso.

L166. Please explain what these acronyms mean.

R: we would like to thank the Reviewer for pointing our mistake, We’ve forgot to remove a parameter didn’t used in the manuscript (ACR). We explained the remaining acronyms in the proper point of the manuscript (namely “Placenta evaluation”).

L188. Please rephrase and correct this.

R: the sentence was modified as suggested by the Reviewer.

L237-238. Please move this explanation to when DWG is first mentioned in the text.

The explanation of this figure is unclear. Please formulate a better one.

R: we reported the acronym explanation in the proper portion of the materials and methods (namely “Puppy management and data collection”). Furthermore, we tried to improve the clearness of the caption. We hope in this manner it could be clearer for the reader.

L246-247. Rephrase this sentence to explain which characteristic of the puppies is comparable with the mean. Is it the PE, the neonatal survival rate or something else?

R: as correctly noted by the Reviewer, the sentence was not clear. We improved the information, hoping this increases the clearness for the reader.

L260. Please explain the meaning of this acronym.

R: We would like to thank the reviewer for the comments. We found that in some points, we used different acronyms for the same parameter. We checked the whole manuscript, uniforming the acronyms we used.

L264. Please rephrase this.

R: the sentence was widely revised to increase its clearness.

L274. Please explain the meaning of this acronym.

R: as for other points, all the acronyms were verified.

L286. Please explain which labyrinth you are writing about.

R: we modified the section mentioned by the reviewer, improving the clearness.

L307. Please ad a section for Conclusions after this one.

R: a conclusion was inserted in the manuscript.

Reviewer 4 Report

Comments and Suggestions for Authors

The manuscript presented by Gloria et al., analyzes the relationship between placental efficiency and survival in Doberman and Boxer puppies. Although the topic is original and interesting, profound changes and a rethinking of the paper are required so that it can be published.

The abstract does not mention that an analysis of the vascularity will be performed

In introduction:

As explained, it seems that the trophectoderm is the only embryonic/fetal component in the placenta. The general introduction to canine placentation is based on appointments of 10 years or more, some aspects must be corrected; For example, currently in carnivores the concept of hematophagous organs and not hematomas is used to refer to the structures specialized in iron uptake. Please rewrite the introduction looking for more up-to-date literature on placentation in carnivores.

In Material and Methods:

 Describe the APGAR method beyond the reference

It is not clear how the authors will analyze the effect of each female. If one female gave birth 10  pups and another 4, variables between those females unrelated to the placenta may affect the results.

The authors claim to analyze the allantochorion, but the placenta released in canine birth has maternal components that are never mentioned.

The placenta removed during the birth of the dog is not histologically homogeneous. It is not clear how the areas were chosen to prevent regional differences from generating non-case-dependent variations in capillary density.

Finally, the use of vascular markers would enrich the work. Currently it does not seem appropriate to analyze the vascularity of an organ such as the placenta using only hematoxylin and eosin.

In the results:

 it would be important to incorporate more tables and graphs to facilitate reading.

Discusision:

Must be rewritten. On the one hand, the aforementioned female effect must be considered; But in addition, it is necessary to incorporate a discussion about the type of placenta since the results are compared with what is known in humans, mice, sheep and pigs; all with placentas that have a different number of layers in the placental barrier than dogs.

General considetation:

The authors mention large breeds and only use two breeds. I consider that the results cannot be generalized to all large breeds and it should be clarified from the title which breeds are being worked with.

Author Response

The abstract does not mention that an analysis of the vascularity will be performed

R: The evaluation of the capillary density was mentioned in the abstract.

In introduction:

As explained, it seems that the trophectoderm is the only embryonic/fetal component in the placenta. The general introduction to canine placentation is based on appointments of 10 years or more, some aspects must be corrected; For example, currently in carnivores the concept of hematophagous organs and not hematomas is used to refer to the structures specialized in iron uptake. Please rewrite the introduction looking for more up-to-date literature on placentation in carnivores.

R: we revised deeply the introduction, reporting the recent references on this matter and reorganizing the sequences of the sentences. We hope we improve the clearness of the section.

In Material and Methods:

Describe the APGAR method beyond the reference

R: The description of the APGAR method used in this study was reported.

It is not clear how the authors will analyze the effect of each female. If one female gave birth 10 pups and another 4, variables between those females unrelated to the placenta may affect the results.

R: We agree with the comment of the reviewer that the number of fetuses in a litter could affect the results obtained for each puppy. The litter size recorded in this study was between 5 to 9. To consider the female effect, we included in the general linear model the litter size as a fixed variable. The litter size was never found significant in the considered parameters. We reported this information in the results section. To verify intra-litter variability, we introduced new variables, calculated as the absolute deviation of each birthweight, chorioallantois weight, and chorioallantois surface, and we used also these variables in the statistical evaluation of the differences between WPE- and SPE-based clusters. We introduced these calculated variables to normalize the values recorded on the mean values of the litter size. Also, in this case, we have found similar values for the absolute deviations in all the parameters between clusters. We reported this information in the results.

The authors claim to analyze the allantochorion, but the placenta released in canine birth has maternal components that are never mentioned.

R: As correctly mentioned by the Reviewer, at expulsion/extraction a maternal component was present with fetal membranes. In the present study, we focused on the chorioallantois to simplify the description of our procedure. We mentioned the presence of maternal components in the first description of the material, claiming our simplification in the following parts of the manuscript.

The placenta removed during the birth of the dog is not histologically homogeneous. It is not clear how the areas were chosen to prevent regional differences from generating non-case-dependent variations in capillary density.

R: Assuming that regional differences in the placental aspects may be present, we selected 3 transversal strips of about 2 cm, in three different regions of the chorioallantois. Three slices from each section were transferred on the slide and analyzed. From each slide, 5 randomly selected fields were digitalized and analyzed. We implemented this description in the materials and methods, to underline our strategies to reduce the “regional” effect.

Finally, the use of vascular markers would enrich the work. Currently it does not seem appropriate to analyze the vascularity of an organ such as the placenta using only hematoxylin and eosin.

R: In this study, the placental vasculature was summarized in the placental capillary density, as described in the literature for the dog (Sarli et al., Animals, 2021). We would point out that the present study was not mainly focused on the histology of the placenta, but the capillary density was used to verify AC with low placental efficiency modified this parameter, reported in the literature. The comment of the reviewer is, however, very interesting, because it is surely possible that a more detailed and extensive evaluation of the placental vasculature, i.e. quantifying the blood bed in this organ, could show evidence of placental adaptation. This could be an interesting point, requiring proper manipulation, preparation and staining of the samples, which could be explored in the future.

In the results:

it would be important to incorporate more tables and graphs to facilitate reading.

R: Some tables were introduced, especially in the descriptive part of the results.

Discusision:

Must be rewritten. On the one hand, the aforementioned female effect must be considered; But in addition, it is necessary to incorporate a discussion about the type of placenta since the results are compared with what is known in humans, mice, sheep and pigs; all with placentas that have a different number of layers in the placental barrier than dogs.

R: We would like to thank the reviewer for the suggestions. We implemented the discussion regarding the effect of litter size on the measurable parameters of the puppy and the placenta. Furthermore, we underlined the fact that differences in vasculature pattern could be related to inter-specific differences.

General considetation:

The authors mention large breeds and only use two breeds. I consider that the results cannot be generalized to all large breeds and it should be clarified from the title which breeds are being worked with.

R: we modified the title of the manuscript, explicating the breeds included in this study.

Round 2

Reviewer 2 Report

Comments and Suggestions for Authors

I would to thank the authors for implementing comments from the reviewers. The paper has been consequently well improved.

Reviewer 3 Report

Comments and Suggestions for Authors

The manuscript has been improved and can be be published after minor revision.

Comments on the Quality of English Language

Minor language and formatting mistakes: in L279 "munutes" is used instead of minutes; the title of table 2 starts with a lowercase letter instead of a capital one; in the explanation to figure 1 the acronym for chorioallantois is A instead of CA.

Reviewer 4 Report

Comments and Suggestions for Authors

The article was rewritten AND the authors realize the suggested changes. The manuscript can be published